# Biomarkers in Localized Prostate Cancer: From Diagnosis to Treatment

**DOI:** 10.3390/ijms26167667

**Published:** 2025-08-08

**Authors:** Marta Lopez-Valcarcel, Fernando Lopez-Campos, Juan Zafra-Martín, Irene Cienfuegos Belmonte, José Daniel Subiela, María Ruiz-Vico, Sandra Fernandez Alonso, Jose Angel Garcia Cuesta, Felipe Couñago

**Affiliations:** 1Department of Radiation Oncology, Puerta de Hierro University Hospital, 28222 Madrid, Spain; mlvalcarcel@salud.madrid.org; 2Department of Radiation Oncology, Ramon y Cajal University Hospital, 28034 Madrid, Spain; flcampos@salud.madrid.org; 3Department of Radiation Oncology, Hospital Universitario Virgen de la Victoria, 29010 Málaga, Spain; 4Department of Urology, Hospital Virgen del Puerto Plasencia, Cáceres, 10600 Plasencia, Spain; irene.cienfuegos@gmail.com; 5Department of Urology, Hospital Universitario Ramón y Cajal, IRYCIS, Universidad de Alcala, 28034 Madrid, Spain; 6Department of Clinical Oncology, Hospital Universitario 12 de Octubre, 28041 Madrid, Spain; 7Department of Radiation Oncology, Hospital Universitario 12 de Octubre, 28041 Madrid, Spain; s.fernandezalo@gmail.com; 8Department of Clinical Oncology, GenesisCare Spain, San Francisco de Asis University Hospital, 28002 Madrid, Spain; 9Department of Radiation Oncology, GenesisCare Spain, Hospital Universitario Vithas La Milagrosa, 28003 Madrid, Spain

**Keywords:** localized prostate cancer, biomarkers, personalized medicine, radiomics

## Abstract

Prostate-specific antigen (PSA) has been the primary biomarker used for the detection and monitoring of prostate cancer for decades. However, its limited specificity and prognostic accuracy have led to the development of novel molecular and imaging biomarkers aimed at improving the clinical characterization of localized disease. This review critically examines recent advances in urinary biomarkers (e.g., PCA3, SelectMDx), tissue-based genomic assays (Oncotype DX Prostate, Prolaris, Decipher), and imaging techniques such as multiparametric magnetic resonance imaging (mpMRI) and prostate-specific membrane antigen positron emission tomography (PET-PSMA). We evaluate their diagnostic performance, prognostic value, and clinical utility in risk stratification and individualized treatment decision-making. Methodological and clinical barriers to their routine implementation are also discussed. Current evidence supports the multidisciplinary integration of these biomarkers to overcome the limitations of PSA, improve biopsy decision-making, better distinguish indolent from aggressive tumors, and optimize therapeutic strategies. Finally, future research directions aimed at validating and incorporating emerging biomarkers into clinical practice are outlined, with the goal of improving outcomes in patients with localized prostate cancer.

## 1. Introduction

Prostate cancer (PCa) is the second most commonly diagnosed malignancy in men and the fifth leading cause of cancer-related death worldwide [1,2]. According to GLOBOCAN 2020, over 1.4 million new prostate cancer cases were reported that year, accounting for nearly 14% of all male malignancies. In high-incidence regions such as North America and Western Europe, the lifetime risk of developing PCa is estimated at 12–13% [3]. These figures underscore the need to optimize diagnostic and therapeutic strategies, particularly in localized disease, where timely intervention is associated with improved clinical outcomes [1,2,4].

Currently, early detection of PCa relies on a combination of clinical parameters and serum prostate-specific antigen (PSA) levels, the only blood-based biomarker widely used in clinical practice [5]. However, PSA has well-documented limitations, particularly its limited sensitivity and specificity, making it difficult to differentiate between benign and malignant processes, as well as between clinically significant and indolent tumors [1]. These limitations have contributed to overdiagnosis and overtreatment in approximately 20–50% of patients undergoing PSA-based screening, especially those with low-risk or indolent tumors, resulting in unnecessary interventions and reduced quality of life [6,7].

In response, several novel biomarkers have been developed to enhance diagnostic accuracy and support more individualized treatment decisions [4]. Blood-, urine-, and tissue-based assays—including PSA derivatives and kallikrein panels—have demonstrated clinical utility in risk stratification. Additional tools, such as gene expression signatures and exosomal RNA profiles, help reduce unnecessary biopsies and support more precise diagnostic strategies [8]. Advances in imaging, particularly multiparametric magnetic resonance imaging (mpMRI) and prostate-specific membrane antigen positron emission tomography (PSMA PET), have improved tumor localization and staging, and are now part of standard diagnostic workflows [9]. In parallel, artificial intelligence (AI) models applied to radiomics and digital pathology are emerging as promising tools to enhance diagnostic precision and guide treatment planning [10].

Biomarkers enable the detection of biological alterations that drive cancer onset, progression, and treatment response. The ideal biomarker should be accessible through minimally invasive and cost-effective methods and demonstrate high sensitivity and specificity. It should also be able to discriminate aggressive tumors from indolent disease [11]. While no single biomarker fulfills all of these criteria, many show significant clinical utility in specific settings.

This review synthesizes validated biomarkers in localized PCa by clinical function, sample type, and indications. The proposed framework is intended to support risk stratification and guide clinical decision-making.

## 2. Methods and Materials

A comprehensive narrative review was conducted to critically integrate current evidence on biomarkers used in the diagnosis, prognosis, and treatment personalization of localized prostate cancer. The relevant literature was identified through a structured search of PubMed, Scopus, the Cochrane Library, Google Scholar, and ScienceDirect, covering studies published between January 2000 and June 2025.

Search terms included “biomarkers,” “PSA,” “imaging modalities,” “genomic classifiers,” “artificial intelligence,” “microRNA,” “lncRNA,” and “SNPs,” in combination with “localized prostate cancer.” Additionally, the reference lists of selected articles were manually reviewed to identify additional pertinent publications.

Studies were included if they met the following criteria: (1) original articles published in English; (2) investigations of biomarkers derived from blood, urine, tissue, or imaging with clinical relevance to localized PCa (including diagnosis, risk stratification, prognosis, or treatment selection); and (3) availability of full text and sufficient methodological detail. The exclusion criteria included preclinical studies, duplicate publications, abstracts without complete data, and articles not published in English.

Two independent reviewers screened titles and abstracts, assessed full texts, and resolved discrepancies through consensus. References were managed using Mendeley Reference Manager v1.19.8 (Elsevier, Amsterdam, The Netherlands). Study selection prioritized clinical relevance, methodological rigor, and translational applicability. Biomarkers with demonstrated clinical utility, prospective validation, or inclusion in evidence-based guidelines were prioritized.

## 3. Classification and Clinical Roles of Biomarkers in Localized Prostate Cancer

The management of localized PCa has evolved from a standardized approach based on PSA and histopathological findings to a more personalized strategy incorporating molecular and imaging biomarkers. To effectively integrate these tools into clinical practice, structured classification is essential. These molecular tools can be functionally grouped into five categories, diagnostic, prognostic, predictive, surrogate, and theranostic, each addressing a specific clinical need [12].

Most assays rely on multivariable algorithms that combine molecular and clinical data to estimate individualized risk. The Prostate Health Index (PHI), for example, is calculated as ([−2] proPSA/free PSA) × √PSA, while the 4Kscore integrates kallikrein markers and clinical factors. Genomic classifiers such as Decipher, Oncotype DX, and Prolaris analyze gene expression signatures to predict tumor aggressiveness or treatment benefit. Although these formulas are often proprietary, their clinical validity has been confirmed in independent studies [4]. Overall, biomarkers derived from blood, urine, tissue, and imaging provide a minimally invasive, integrated framework for individualized risk classification.

### 3.1. Diagnostic Biomarkers

These biomarkers improve cancer detection in patients with PSA levels between 2 and 10 ng/mL or ambiguous clinical findings [13]. The PHI, which combines total, free, and [−2] proPSA, enhances specificity for clinically significant PCa [14]. The 4Kscore measures total, free, intact PSA, and hK2, and incorporates clinical variables to estimate the risk of high-grade disease. It is useful in both biopsy-naïve patients and those with previously negative biopsies [15].

Urinary assays like PCA3 detect overexpression of the prostate cancer gene 3, a non-coding RNA highly specific for prostate cancer, and are primarily used after a negative biopsy [16]. SelectMDx evaluates urinary expression of *HOXC6* and *DLX1*, genes associated with aggressive tumors, to identify clinically significant disease and reduce unnecessary biopsies, especially valuable in patients with PIRADS ≤ 3 or a prior negative biopsy [17]. ExoDx analyzes exosomal RNA (ERG, PCA3, SPDEF) in urine without requiring a digital rectal exam and provides a risk score for high-grade prostate cancer [18]. The Stockholm3 (S3M) model combines plasma biomarkers (PSA, hK2, MSMB, MIC1), SNPs, and clinical data into a multivariable algorithm that outperforms PSA alone in predicting clinically significant cancer [19]. These tools are frequently used alongside multiparametric MRI (mpMRI) to optimize the selection of candidates for initial or repeat biopsy [19,20,21,22].

### 3.2. Prognostic Biomarkers

These biomarkers assess disease aggressiveness independently of treatment and help estimate the risk of progression, recurrence, or metastasis. They support key clinical decisions such as active surveillance versus curative treatment, particularly in men with low- or intermediate-risk disease.

Tissue-based genomic classifiers include Decipher, a 22-gene expression signature that generates a metastasis risk score independently of traditional clinical variables [23]. Oncotype DX GPS analyzes the expression of 17 genes across four biological pathways to produce a genomic prostate score that predicts adverse pathology and biochemical recurrence [24,25]. Prolaris quantifies the expression of 31 cell cycle progression genes to calculate a cell cycle progression (CCP) score, which correlates with tumor aggressiveness and long-term outcomes [26].

ProMark is a protein-based assay using quantitative immunofluorescence to evaluate eight protein biomarkers, particularly useful in patients with Gleason score 3+3 or 3+4 disease [27].

Non-invasive imaging parameters such as low ADC on mpMRI [28] and high SUVmax on PSMA PET [9] are being investigated as potential prognostic biomarkers, given their association with higher tumor grade and recurrence risk, although further validation is required.

### 3.3. Predictive Biomarkers

These biomarkers indicate the likelihood of benefit from a specific therapy, allowing treatment to be tailored according to tumor sensitivity or resistance. Unlike prognostic biomarkers, which reflect disease progression, predictive biomarkers guide therapeutic selection.

Decipher has been shown to aid in selecting adjuvant radiotherapy after prostatectomy by identifying patients at a higher risk of recurrence who may benefit from treatment intensification [29]. PORTOS is a genomic signature derived from DNA damage response genes that may help identify patients more likely to benefit from dose-escalated radiotherapy [30]. Oncotype DX GPS, in addition to its prognostic role, has shown predictive value for outcomes following surgery or radiation therapy [24].

AR-V7, a splice variant of the androgen receptor detected in circulating tumor cells (CTCs), predicts resistance to androgen receptor signaling inhibitors such as enzalutamide and abiraterone. It thus serves as a predictive biomarker of poor response. Its presence also correlates with more aggressive disease and poorer clinical outcomes, offering additional prognostic information [31,32,33].

### 3.4. Surrogate Biomarkers

Surrogate biomarkers act as early indicators of treatment efficacy and are often used as intermediate endpoints in clinical trials or for longitudinal monitoring.

PSA kinetics—including PSA doubling time, PSA nadir, and time to biochemical recurrence—are widely used to monitor response after surgery or radiotherapy. However, their correlation with survival outcomes remains inconsistent [34].

Imaging-based parameters such as ADC values from mpMRI and SUVmax from PSMA PET are being evaluated as quantitative surrogate biomarkers to detect early treatment response or disease progression. Prospective validation is needed before their full integration into clinical decision-making [9,35].

### 3.5. Theranostic Biomarkers

These biomarkers integrate diagnostic and therapeutic functions. PSMA is the most widely studied theranostic target. PSMA PET imaging with ^68^Ga- or ^18^F-labeled tracers allows for accurate disease localization and plays a validated role in guiding radioligand therapy (e.g., ^177^Lu-PSMA) in metastatic settings.

In localized prostate cancer, PSMA PET is increasingly used for improved staging, detection of extracapsular extension, and focal therapy planning. While its therapeutic use in localized disease remains investigational, PSMA PET yields biological insights that inform risk-adapted strategies and the development of biologically informed treatment strategies [36].

A clear understanding of this classification is essential for the implementation of precision oncology in localized prostate cancer. It facilitates the integration of molecular biomarkers into routine clinical practice and enables personalized treatment strategies based on tumor biology and individual patient risk. Figure 1 and Table 1 provide a visual summary of this conceptual framework and illustrate its relevance across key decision-making points.

Sequential use of blood, urine, tissue, and imaging biomarkers enhances biopsy selection, risk stratification, and treatment planning across key decision points.

Biomarkers are grouped by diagnostic, prognostic, predictive, surrogate, or theranostic roles, with examples, sample types, clinical applications, and interpretation criteria. 

## 4. Diagnosis: How to Select Patients Who Really Need a Biopsy?

Population-based PSA screening for PCa has been shown to reduce mortality by 27% and the incidence of metastasis by 33%, according to the European Randomized Study of Screening for Prostate Cancer (ERSPC) [13]. However, its low specificity (especially in ranges between 2.5 and 10 ng/mL) is associated with a high rate of unnecessary biopsies. To improve patient selection, the European Association of Urology (EAU) recommends the use of multiparametric magnetic resonance imaging (mpMRI) in men with PSA levels between 3 and 20 ng/mL due to its high negative predictive value (NPV), although its positive predictive value (PPV) remains variable [38].

In this context, new non-invasive biomarkers are being developed to refine biopsy indications.

The Prostate Health Index (PHI) demonstrated a sensitivity of 82% and a specificity of 84% in a cohort of 421 patients, with a cutoff of 43 and an area under the curve (AUC) of 0.77. This was superior to total PSA (AUC 0.58) and the free PSA/total PSA ratio (AUC 0.64) [39]. In combination with mpMRI, the PHI improves detection of clinically relevant tumors even in PI-RADS ≤ 3 lesions. It avoids 39.5% of unnecessary biopsies while maintaining a 97% detection rate for significant cancers [28].

The 4Kscore test reached a sensitivity of 93% and a specificity of 55% in patients with PSA levels between 2 and 10 ng/mL [40]. In the ProScreen study, a threshold <7.5% avoided the need for mpMRI in 19% of patients with PSA ≥ 3 ng/mL, resulting in a 28% reduction in biopsies and a 41% overall decrease in mpMRI use [41].

The Stockholm3 (S3M) test reduced unnecessary biopsies by 34% with triple specificity compared to PSA, while maintaining sensitivity for high-risk tumors in a cohort of 59,159 patients [19]. Furthermore, in a multiethnic population (*n* = 2129), the test achieved a 45% reduction in biopsies (between 42 and 52%, depending on ethnic group) without compromising the detection of clinically significant cancers [42].

Among urinary biomarkers, SelectMDx has an NPV of 98% for aggressive tumors with an AUC of 0.76 [43]. With a cutoff value of −2.8, it reported a sensitivity of 90% and a specificity of 50% [43]. It avoids 38% of biopsies but fails to diagnose 10% of high-risk tumors. In combination with mpMRI, it avoids 49% of biopsies and only misses 4.9% of aggressive cancers. Although its performance declines in low-risk populations, it remains useful in cases of previously negative biopsies or initial clinical suspicion [44].

PCA3 is a non-coding RNA overexpressed in prostate cancer approved by the Food and Drug Administration (FDA). With a threshold of 25, it showed a sensitivity of 77.5% and a specificity of 51.1% with an AUC of 0.73, superior to the free PSA/total PSA ratio (AUC 0.66) [45]. This test is adapted and validated for patients with previously negative biopsies, persistently high PSA, clinical suspicion with normal PSA, or active surveillance (AS). Together with clinical variables, it can avoid between 40% and 67% of unnecessary biopsies [46]. However, its PPV is low (33.6%) and it may also be positive in up to 10% of non-invasive lesions [47]. The widespread use of mpMRI has relegated PCA3 to a complementary role. It is mainly used in research but is not recommended as a screening tool [48]. The Michigan Prostate Score (MiPS), which combines PCA3, *TMPRSS2/ERG,* and PSA, improves detection of high-risk tumors and spares mpMRI or biopsy in 35–51% of cases [49].

The ExoDx test, which evaluates exosomal RNA from ERG, PCA3, and SPDEF, showed 92% sensitivity, a 91% NPV, and a 36% PPV in a cohort of 1000 patients with PSA between 2 and 10 ng/mL [50,51]. A threshold of 15.6 avoided more than 25% of biopsies and correlated with histopathological characteristics after radical prostatectomy, demonstrating its usefulness in selecting candidates for AS [22].

When a prostate biopsy is negative but clinical suspicion remains, the ConfirmMDx test can be employed. This assay detects the hypermethylation of the APC, RASSF1, and GSTP1 genes [52]. This alteration reflects the “halo effect”, whereby epigenetic changes in healthy tissue indicate proximity to a malignant area. In the MATLOC study (*n* = 498), ConfirmMDx showed a 90% NPV. In the DOCUMENT study (*n* = 350), the NPV was 88% (95% CI, 85–91%) after two years of follow-up [53]. Subsequently, the EpiScore algorithm improved the performance to an NPV of 96%, which is superior to clinical variables such as PSA, high-grade PIN, or rectal exam [43]. The test is currently included in the National Comprehensive Cancer Network (NCCN) guidelines, although its use is limited by cost and sample availability.

In summary, the PHI and 4Kscore are accessible and validated for use in biopsy-naïve men, while urinary and epigenetic tests such as SelectMDx, ExoDx, and ConfirmMDx offer additional value after negative biopsy or inconclusive mpMRI. The combination of molecular biomarkers and imaging provides the highest diagnostic accuracy, although factors such as availability, cost, and validation across populations should also be considered in clinical decision-making.

## 5. Prognosis: How to Distinguish Between Indolent and Aggressive Tumors?

The management of PCa has evolved beyond the traditional approach based solely on PSA, Gleason score, and clinical stage, particularly for intermediate-risk patients. In this context, genomic tissue classifiers offer valuable tools for improving prognostic stratification and guiding personalized therapeutic decisions [54]. Despite the use of PSA kinetics for AS, its limited discriminatory power has driven the development of more accurate genomic tests [55,56].

Decipher can be applied to both biopsy and prostatectomy specimens, depending on the clinical context and stage of decision-making, by analyzing the expression of 22 genes associated with androgen signaling, proliferation, and immune response [57]. In favorable intermediate-risk patients, high scores (>0.6) were associated with a worse prognosis (*p* < 0.001) and a shorter time to treatment during AS (*p* < 0.001) [58]. Its clinical utility has been evaluated in several studies. In the PRO-ACT trial, it led to treatment intensification of high-risk patients (*p* < 0.001) [59]. In the analysis by Badani et al. [60], it increased the recommendation for AS in low-risk (20%) and high-risk (16%) patients. Finally, the Surveillance, Epidemiology and End Results (SEER) analysis increased the AS rates from 37% to 39% (*p* < 0.001) and reported an association between high scores and advanced stages or greater tumor aggressiveness [61].

OncotypeDx, recommended by the American Society of Clinical Oncology (ASCO) and the NCCN, evaluates 17 genes using reverse transcription polymerase chain reaction (RT-PCR) and generates a score from 0 to 100. This test predicts the likelihood of metastases, mortality, and adverse pathological features (Gleason ≥ 4+3 or pT3+) [62]. In a cohort of 514 patients (91% Caucasian), each 20-point increase was associated with an increased risk of high-grade disease (OR = 2.3; 95% CI: 1.5–3.7) and disease not confined to the prostate (OR = 1.9; 95% CI: 1.3–3) [25]. A meta-analysis of 732 patients showed that the combination of OncotypeDx, the Cancer of the Prostate Risk Assessment (CAPRA) (AUC 0.68–0.73), and the NCCN classification (AUC 0.64–0.70) improved the prediction of adverse pathology, especially in racially diverse cohorts (*n* = 431; 20% African American), with an improvement in AUC from 0.63 to 0.72 [63]. These data support its incremental value when integrated with clinical tools.

Prolaris, also recommended by the ASCO and NCCN, evaluates 46 genes related to cell cycle progression using RT-PCR and generates a 10-point scale to estimate the risk of mortality and metastasis at 10 years [64]. In patients who underwent transurethral resection of the prostate (TURP), a higher risk of PCa mortality was observed (HR = 2.56; *p* < 0.0001). When combined with clinical variables, the AUC improved from 0.80 to 0.88 [65]. In the PROCEDE-1000 study (*n* = 1026), 47.8% of patients had their treatment modified due to the test, with a 72.1% de-intensification and a 26.9% intensification [66]. In a multicenter cohort of 585 men with biopsy-confirmed low-risk prostate cancer, combining the Clinical Cell Cycle Risk (CCR) score with the NCCN criteria increased the proportion of patients eligible for active surveillance from 42.6% to 68.8%. Among those classified as low risk by the CCR, the predicted 10-year prostate cancer-specific mortality was below 2%, with no prostate cancer-related deaths observed during follow-up [67]. In a retrospective series of 3208 patients, the choice for AS doubled, with durability after 3 years that was 1.5 times greater (*p* < 0.0001) [68].

ProMark is a prognostic test that evaluates the expression of eight proteins in prostate biopsies using quantitative immunohistochemistry. It aids decision-making between AS and treatment in patients with Gleason 3+3 or 3+4. In a study by Blume-Jensen et al. [69] (*n* = 381) (AUC 0.78), a score > 0.8 was associated with a sixfold higher risk of progression, whereas a low Gleason 3+3 score showed an NPV of 84%. This test does not require genomic techniques or large tissue samples, facilitating use in resource-limited settings. However, it does not predict the risk of metastasis, and its clinical adoption remains limited [27].

A systematic review by Trabiz et al. [70] concluded that these tests improve the estimation of tumor aggressiveness, with bidirectional risk reclassifications in patients with intermediate-risk PCa and with variations depending on race. Although observational studies show a tendency towards increased AS, randomized trials continue to favor definitive treatments. Overall, these tools have the potential to refine the prognostic classification, but more controlled prospective studies are required to evaluate their clinical impact and cost-effectiveness.

mpMRI in combination with radiomic analysis is emerging as a non-invasive prognostic tool. In low-risk and intermediate-risk patients, a reduction in T2 was associated with a lower PSA level after one year [71]. Furthermore, low apparent diffusion coefficient (ADC) values were associated with a higher risk of recurrence (*p* = 0.002) and progression within five years (*p* < 0.001) [35]. These findings support the value of functional imaging in estimating aggressiveness and the risk of progression.

Positron emission tomography (PET) with ^68^Ga-PSMA-11 is now the preferred method for staging intermediate and high-risk PCa over computed tomography and bone scans, and it also provides quantitative parameters such as SUVmax. An intraprostatic SUVmax>8 has been consistently associated with a high Gleason score (*p* < 0.05), positive margins (*p* < 0.01), an advanced stage, and shorter biochemical progression-free survival. Even in cases of Gleason ≤ 3+4, high SUVmax indicates a higher risk of recurrence, which further supports its value in reclassifying candidates for AS or local treatment [9,72]. These data suggest that SUVmax could be an emerging prognostic biomarker in localized disease.

Altogether, these genomic and imaging-based biomarkers enhance conventional risk stratification in localized PCa and support more individualized and evidence-based prognostic assessments.

## 6. Treatment: Should Treatment Be Intensified or De-Intensified?

PSA remains essential for the stratification and management of PCa. A pretreatment PSA value of >20 ng/mL indicates a high risk of recurrence [73,74]. After treatment, parameters such as a PSA doubling time < 6–12 months [75,76], persistence of a high PSA level (≥0.2 ng/mL after prostatectomy) [76], or early biochemical recurrence (<18–24 months) [77] are suggestive of an aggressive disease and justify hormonal intensification. Conversely, a PSA nadir < 0.1 ng/mL after radiotherapy or late recurrence (>24 months) may benefit from delaying or reducing androgen deprivation therapy (ADT) [78,79]. Integrating PSA with other tools, such as CAPRA and the NCCN, in risk groups improves prognostic stratification [80].

The AR-V7 splice variant of the androgen receptor is associated with resistance to androgen receptor pathway inhibitors (ARPIs) and adverse clinical outcomes. Although it is generally absent in untreated localized prostate cancer, AR-V7 is detected in 13–20% of patients after prostatectomy and subsequent androgen deprivation therapy (ADT), particularly in high-risk cases. Its presence correlates with significantly reduced biochemical progression-free survival (bPFS), overall survival (OS), and metastasis-free survival (mPFS) [31]. AR-V7 can be identified using clinically available methods such as RT-PCR, RNA sequencing of prostate tissue, immunohistochemistry, and non-invasive liquid biopsy via RT-PCR in circulating tumor cells. Patients positive for AR-V7 have a median bPFS of around 11 months, compared to over 70 months for those who test negative. This suggests that the variant drives tumor progression despite ADT and limits the efficacy of conventional hormonal therapies [31]. These findings support the use of AR-V7 testing to improve risk stratification and guide treatment decisions in locally advanced prostate cancer. Patients harboring AR-V7 may benefit more from early chemotherapy or enrollment in clinical trials. Incorporating AR-V7 into clinical workflows may enhance therapeutic personalization and inform early treatment intensification.

Among the genomic tools, Prolaris can estimate the risk of metastasis in localized PCa. In a prospective cohort of 554 patients, the combined cell cycle risk (CCR) score demonstrated significant prognostic value after three years (AUC 0.74; *p* = 0.001). In patients with a high CCR, the metastasis rate was 14% with single-agent treatment compared to 3% with multimodal treatment [81]. Furthermore, in a cohort of 56,485 patients, the benefit of adding ADT to radiotherapy varied according to CCR, with a 17.1% reduction in risk at ten years for CCR = 3.69 [82].

The genomic prostate score or GPS (Oncotype) also predicts the response to radiotherapy. A 20-point increase in the GPS was associated with a higher risk of biochemical failure (HR: 3.62; CI 95%: 2.59–5.02), distant metastasis (HR: 4.48; CI 95%: 2.75–7.38), and death from PCa (HR: 5.36; CI 95%: 3.06–9.76). In patients with a GPS > 40, the risk of metastasis (HR: 5.22; CI 95%: 3.72–7.31), biochemical recurrence (HR: 4.41; CI 95%: 2.29–8.49), and mortality from PCa (HR: 3.81; CI 95%: 1.74–8.33) was significantly higher [24], with no differences according to race (*p* = 0.923) [83].

The genomic classifier Decipher can predict metastatic dissemination and biochemical recurrence after radiotherapy. In the phase III RTOG 9601 trial, patients with a score > 0.6 who were treated with radiotherapy plus bicalutamide had a higher 12-year OS rate (70% vs. 51%, *p* = 0.005), although this benefit was limited to those with a high genomic risk [29]. In the NRG/RTOG 0126 substudy, only patients with high scores benefited from the higher dose [30]. The ongoing phase III GUIDANCE trial (NRG-GU010) is evaluating the use of Decipher in unfavorable intermediate-risk PCa to inform decisions about treatment, including radiotherapy alone, radiotherapy plus short-course ADT, or radiotherapy plus ADT plus darolutamide [84].

PORTOS is a 24-gene signature derived from Decipher that predicts the response to radiotherapy. It has been validated in the SAKK 09/10 and RTOG 0126 trials, identifying patients who could benefit from dose escalation. In the SAKK 09/10 trial, patients with high scores benefited from receiving 70 Gy rather than 64 Gy. In RTOG 0126, patients in the higher tertile achieved better results with 79.2 Gy versus 70.2 Gy. At a molecular level, the genes in PORTOS are associated with hypoxia and immunological pathways [30].

In the postoperative context, Decipher can also inform decisions regarding adjuvant or salvage therapy choices [85]. In the PRO-IMPACT study, treatment was modified in 30% of cases [84]. In the G-MINOR trial, high scores were associated with the need for adjuvant radiotherapy (*p* = 0.009) [86]. The SPPORT study found that the addition of nodal radiotherapy and ADT in patients with scores > 0.6 reduced the risk of progression (*p* < 0.001), with an absolute benefit of 27% at ten years (HR: 0.60; CI 95%: 0.37–0.97; *p* = 0.04) [87]. Finally, in phase II of the STREAM trial, despite the intensification with enzalutamide, ADT, and salvage radiotherapy, almost 50% of high-risk patients experienced recurrence within three years. Patients with a differentiated luminal subtype achieved PFS of 89% compared to 19% for those with proliferating luminal subtype. A loss of PTEN (HR: 1.32; *p* = 0.01) and homologous recombination deficiency (HRD) (HR: 1.21; *p* = 0.009) were associated with a worse prognosis. In contrast, a good response to ADT (HR: 0.75; *p* = 0.01) predicted better outcomes [88].

Overall, genomic and molecular biomarkers are emerging as key tools for tailoring treatment intensity in localized and locally advanced PCa, providing complementary information to guide therapy beyond conventional parameters.

## 7. Artificial Intelligence Applications in Prostate Cancer Radiomics

Artificial intelligence (AI), particularly through radiomics and machine learning (ML), is increasingly being investigated to support diagnosis, risk stratification, and treatment planning in localized PCa. Radiomics enables the extraction of quantitative features from images such as T2-weighted MRI, ADC maps, and PSMA PET/CT, capturing tissue heterogeneity and microstructural complexity beyond visual interpretation [10]. One of its key applications is the differentiation of clinically significant prostate cancer (csPCa), especially in ambiguous lesions like PI-RADS 3. Radiomic models based on mpMRI or biparametric MRI have shown high predictive accuracy, with AUCs between 0.86 and 0.98—surpassing clinical scores (AUC 0.79) [89,90].

Texture features such as entropy, short-rung emphasis, and uniformity from ADC or T2 images are strongly associated with Gleason ≥ 7. Combined models that integrate radiomics, PSA density, and PI-RADS category can reduce unnecessary biopsies by approximately 20% without compromising sensitivity [91].

Radiomics also aids prognostic stratification. Tools such as RadS or RadClip have shown utility in predicting biochemical recurrence (BCR) after prostatectomy, with HRs exceeding 7 and C-indices up to 0.77. Specific features, including Small-Zone Emphasis from Gray-Leve Size Zone Matrix (SZEGLSZM), independently predict BCR (AUC > 0.82), correlating with tumor cellularity and diffusion restriction [90,92].

However, important limitations persist, including protocol variability, overfitting, and limited external validation. Multimodal models combining radiomics, genomics (e.g., RNA profiles, SNPs), and clinical data further improve prediction of adverse pathology, with AUCs ≥ 0.88 [93,94].

For treatment monitoring, radiomics has been used to identify early biochemical response and radiation-induced toxicity. Texture-based features from MRI can predict response to ADT (AUC 0.81–0.86) [95], while CT-based models have been developed to predict radiation-induced cystitis or proctitis (AUCs 0.77 and 0.71, respectively) [96]. In PSMA PET/CT, radiomics models have outperformed conventional visual interpretation in the detection of subclinical disease. Features such as SUV entropy, surface-to-volume ratio, or tumor-to-background ratio achieve an AUC ≥ 0.93 with a sensitivity > 80% [97] and also show high accuracy in detecting nodal metastases (AUC 0.95) [98].

In digital pathology, DL-based platforms are also emerging. ArteraAI is a multimodal deep learning model that analyzes digitized H&E-stained biopsies to predict the individual benefit of adding ADT to radiotherapy. In a phase III validation cohort (*n* = 1594), it identified patients with a 15-year distant metastasis risk reduction from 14.4% to 4.0% (HR 0.34) when ADT was added [92].

In parallel, PATHOMIQ_PRAD is an AI-powered histologic classifier that integrates morphologic and subvisual features from H&E-stained whole-slide images to predict recurrence and metastasis after radical prostatectomy. In a blinded validation study involving 344 patients, PATHOMIQ_PRAD demonstrated strong prognostic accuracy, with hazard ratios of 3.27 for BCR and 10.10 for distant metastases, outperforming standard clinical and genomic classifiers [99].

Despite encouraging results, most current studies remain retrospective and single-center. Manual segmentation, a lack of standardization, and protocol heterogeneity limit reproducibility. Future efforts should focus on prospective multicenter validation and the adoption of standardized imaging and analysis workflows. Properly validated AI tools may ultimately support risk-adapted, biologically informed decision-making in localized PCa—particularly in borderline scenarios such as PI-RADS 3 lesions, low-risk tumors under surveillance, or intermediate-risk disease where radiogenomic signatures may guide the indication for radiotherapy or systemic therapy.

## 8. Discussion

Integrating biomarkers into the management of localized PCa is transforming diagnosis, prognostic stratification, and decision-making in PCa by promoting a more precise and personalized approach [100]. Although PSA remains the reference biomarker due to its availability and accessibility, its limitations in terms of sensitivity and specificity highlight the need to incorporate more advanced molecular tools [101]. The classification of biomarkers into diagnostic, prognostic, predictive, surrogate, and theranostic categories provides a functional and clinically actionable framework for their use across different stages of care.

In the diagnostic setting, blood- and urine-based tests such as the PHI, the 4Kscore, SelectMDx, PCA3, and circulating tumor cells offer greater specificity than PSA alone, particularly in men with PSA levels between 2 and 10 ng/mL [20,46,51,102,103,104,105].

When combined with mpMRI, these tools can reduce unnecessary biopsies by up to 50% while maintaining high sensitivity for clinically significant prostate cancer [28,41,44]. Multimodal liquid biopsy panels, such as Stockholm3 and MiPS, integrate genomic, proteomic, and clinical data to enhance diagnostic accuracy, particularly in biopsy-naïve patients or those with prior negative results [19,42,49].

In terms of prognosis, genomic classifiers such as Decipher, OncotypeDx, and Prolaris enable more accurate predictions of tumor aggressiveness, recurrence risk, and metastatic potential, often reclassifying 30–50% of patients [4,64].

These tools are particularly valuable when combined with histopathology and mpMRI. Imaging-derived biomarkers, such as low ADC values on mpMRI and high SUVmax (>8) on PSMA PET, are also emerging as non-invasive indicators of adverse pathology and early biochemical recurrence [9,35,72].

Beyond prognosis, several biomarkers demonstrate predictive value. Tools such as Decipher, PORTOS, Oncotype GPS, and AR-V7 help identify patients most likely to benefit from radiotherapy, ADT, systemic intensification [30,31,32,33,81,82,83]. The PORTOS signature informs radiosensitivity and dose escalation decisions [30], while AR-V7 is associated with resistance to AR-targeted therapies and poor outcomes [31,33]. These biomarkers allow for escalation in high-risk patients and support treatment de-intensification in favorable-risk cases.

In parallel, AI has emerged as a promising tool for integrating radiomic, genomic, and histopathologic data. Quantitative imaging features extracted from mpMRI and PSMA PET can predict biochemical recurrence and treatment response, with AUC values frequently exceeding 0.80 in independent validation cohorts [91,106,107].

Predictive models such as RadClip, RadS, and ArteraAI have shown the potential to outperform standard nomograms like CAPRA or Decipher in selected settings [106,107]. However, broad clinical adoption of AI tools requires external validation, standardized imaging protocols, and regulatory approval [82,92,108,109,110].

Importantly, several biomarkers are supported by prospective and real-world evidence. Genomic classifiers such as Decipher and Prolaris have demonstrated prognostic and predictive utility in large observational cohorts and randomized trials, including PROCEDE-1000 [66], RTOG 9601 [29], SAKK 09/10 [30], STREAM [88], and the ongoing GUIDANCE trial. Their inclusion in leading clinical guidelines (e.g., NCCN, ASCO, EAU) further supports their implementation in routine care.

Despite their promise, significant challenges remain. Barriers to widespread adoption include high costs, limited access, clinician unfamiliarity, and variability in healthcare infrastructure. Much of the current evidence is based on retrospective or single-center studies. More prospective, multicenter, randomized trials are needed to evaluate long-term outcomes such as metastasis-free survival and prostate cancer-specific mortality [70].

Additionally, health-economic analyses will be essential to support cost-effectiveness and equitable integration into clinical workflows [108,111,112].

To facilitate implementation, Table 2 summarizes validated biomarkers in localized PCa, detailing sample type, diagnostic performance, interpretation, and validated use. Figure 2 proposes a biomarker-informed therapeutic algorithm for localized PCa, encompassing both initial diagnosis pathways and post-treatment recurrence.

Finally, the successful implementation of biomarkers and AI-based tools requires overcoming technical and systemic challenges, including protocol standardization, staff training, interoperability between technological platforms, and clinical validation in diverse populations. The increasing use of genomic data and AI-based tools also raises ethical, legal, and privacy concerns that must be addressed through robust regulatory frameworks and data protection strategies [109].

In summary, the integration of validated biomarkers, advanced imaging modalities, and AI-based tools is progressively reshaping the clinical management of localized prostate cancer. By enabling biologically informed, risk-adapted, and patient-centered decision-making, these innovations contribute to a more precise and individualized approach to care. As the body of supporting evidence expands and implementation challenges are systematically addressed, such tools are expected to become essential components of routine clinical practice.

## 9. Future Perspectives

Next-generation biomarkers improve diagnostic accuracy, prognosis evaluation, and the decision-making process by identifying critical changes in molecular signaling pathways, thus enabling personalized oncology care. MicroRNA (miR) can be detected in blood and urine, either freely or as part of extracellular vesicles (EVs). They have the potential to serve as non-invasive biomarkers [115]. A combined urinary panel incorporating PSA (miR-572, -1290, -141, -145; -21, -204, and -375) can distinguish between benign and malignant diseases with an AUC ranging from 0.70 to 0.86 [116]. miRs such as miR-145 and let-7a-5p are associated with high-grade tumors (Gleason > 8; AUC 0.68) [117].

Other miRNAs, such as miR-19b and miR-16, demonstrate high sensitivity and specificity: 100% and 95%, and 93% and 79%, respectively [118]. miR-155 is overexpressed in tumor tissue and correlates with PSA, TNM, and tumor volume (*p* < 0.05) [119]. Several prognostic miRNAs have been identified. For instance, miR-2909 can distinguish cancer from benign hyperplasia [120], while miR-34b/c and miR-23a-3p are present in aggressive phenotypes. Conversely, let-7b-5p, miR-128a-3p, -188-5p, -224-5p, and -23b-3p are associated with favorable prognosis [121]. MiR-148a-3p [122] and miR-582-5p [123] can predict biochemical recurrence, progression, and bone metastases.

Long non-coding RNAs (lncRNA > 200 nt) are also emerging as diagnostic and prognostic biomarkers [124]. MALAT1, MAGI2-AS3, PVT1, NEAT1, and CAT2064 improve the diagnostic performance with an AUC of 0.67–0.95 [125,126,127,128,129]. PCAT-1 is related to progression and a high Gleason score (*p* = 0.01) [130], whereas UCA1 (*p* < 0.0001) [131], ZEB1-AS1, and SNHG9 are associated with poor prognosis and metastasis [132]. Conversely, MAGI2-AS3 and PCAT14 are linked to favorable outcomes [133]. lncRNA-based models outperform traditional nomograms in predicting 5-year recurrence, and their detection with droplet digital PCR (ddPCR) in blood and urine is non-invasive [124].

Single-nucleotide polymorphisms (SNPs) are also useful for predicting prognosis [134]. For example, the allele rs6983267 (8q24) is associated with an increased risk of PCa (*p* = 3.4 × 10^−5^) [135], and the allele rs1042522 (TP53) is associated with a Gleason score ≥ 8 (*p* < 0.0001) [136]. SNPs such as rs1400633 (MSH2) can predict a better response to ADT (*p* = 0.002) [137], whereas rs4648302 (PTGS2) is associated with a lower risk of recurrence after prostatectomy (*p* = 0.046) [138]. Rare variants like rs188140481 (frequency 1.6%) confer a markedly increased PCa risk (*p* < 0.001) [139].

The gut microbiota has emerged as a promising source of non-invasive diagnostic and prognostic biomarkers in prostate cancer. Multiple studies have reported significant compositional differences between patients and healthy controls, including shifts in microbial diversity and the relative abundance of specific bacterial taxa. These changes may affect systemic inflammation, hormonal metabolism, and immune modulation—mechanisms that are potentially involved in prostate carcinogenesis. Furthermore, certain microbial signatures have been correlated with disease stage, tumor aggressiveness, and treatment response, highlighting their potential role in risk stratification and therapy monitoring [140]. While these findings are encouraging, further research is necessary to validate microbial biomarkers in larger and more diverse populations, and to clarify causal relationships rather than mere associations. Integrating gut microbiome profiling into clinical practice will require standardized sampling protocols, robust bioinformatic pipelines, and regulatory approval to ensure reproducibility and clinical utility [101,102].

These technological advances are expected to enhance risk stratification and therapeutic decision-making in localized prostate cancer, but further validation and standardization are required for their clinical implementation.

## 10. Conclusions

Validated biomarkers, advanced medical imaging, and AI-based tools are transforming the management of localized prostate cancer by enabling more accurate diagnosis, risk-adapted treatment, and individualized follow-up. Although widespread implementation still faces logistical and regulatory barriers, expanding clinical evidence supports their incorporation into routine practice. As these approaches become more systematically integrated, they are expected to support more effective and biologically guided patient care.

## Figures and Tables

**Figure 1 ijms-26-07667-f001:**
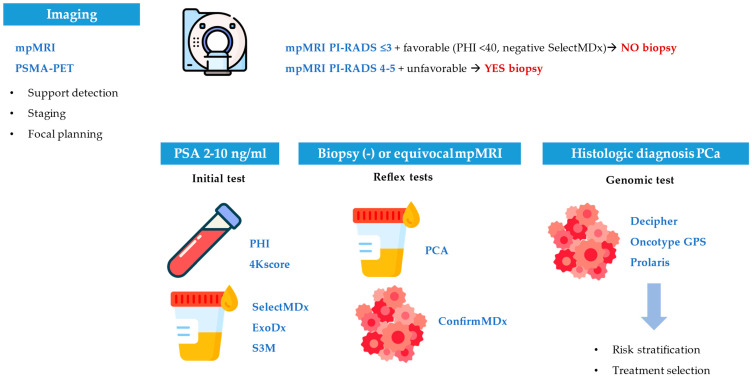
Clinical integration of biomarkers in localized prostate cancer.

**Figure 2 ijms-26-07667-f002:**
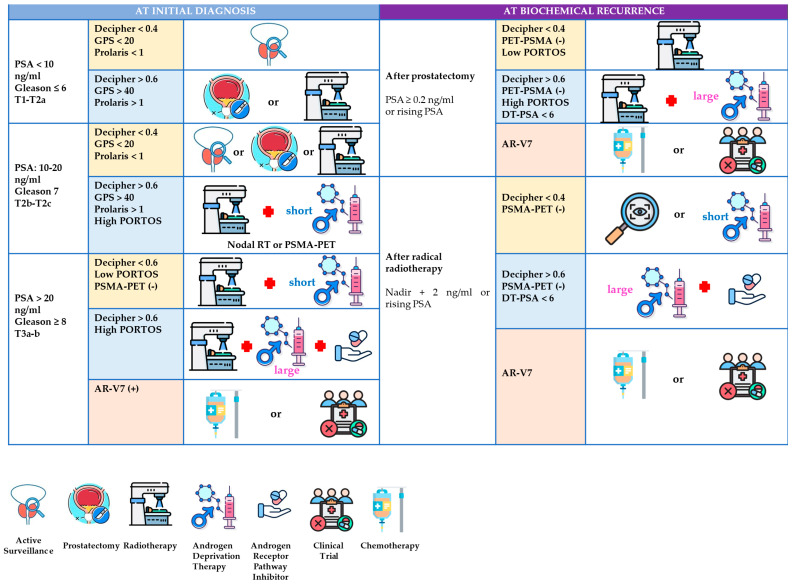
A biomarker-informed therapeutic algorithm for localized prostate cancer: diagnosis and biochemical recurrence. The proposed clinical algorithm integrating biomarkers in localized prostate cancer at diagnosis and after biochemical recurrence. The figure outlines a stratified therapeutic approach based on risk group (low, intermediate, high) and integrates validated biomarkers—such as Decipher, GPS, Prolaris, PHI, and PSMA PET—to guide decisions regarding active surveillance, radiotherapy, androgen deprivation therapy (ADT), and systemic intensification. In the setting of biochemical recurrence, biomarker results (e.g., Decipher, AR-V7, PSMA PET) are used to tailor salvage strategies. This algorithm does not depict experimental or emerging tools not yet implemented in routine clinical practice.

**Table 1 ijms-26-07667-t001:** Classification of biomarkers by clinical function.

Type	Brief Description	Examples	Sample Type	Clinical Use	Validated Context	Calculation and Clinical Value
Diagnostic	Identify clinically significant cancer and reduce unnecessary biopsies	PHI [14], 4Kscore [15], PCA3 [16], SelectMDx [17], ExoDx [18], S3M [19]	Blood, urine	Biopsy decision in men with elevated PSA	Initial or repeat biopsy, PSA 2–10 ng/ml	PHI: ([−2] proPSA/free PSA) × √PSA 4Kscore: % risk of high-grade cancer PCA3: PCA3/PSA mRNA ratio SelectMDx: HOXC6/DLX1 mRNA + clinical data ExoDx: exosomal RNA signature. S3M: algorithm combining SNPSs, proteins, and clinical data
Prognostic	Estimate risk of progression, metastasis, or mortality regardless of treatment	Decipher [23], Prolaris [26], Oncotype DX GPS [24,25], ProMark [27], mpMRI [28]	Tissue, imaging	Risk stratification, active surveillance vs. treatment	Low/intermediate-risk disease	Decipher: metastasis risk score Prolaris: CCP score GPS: genomic score ProMark: 8-protein panel mpMRI: low ADC linked to higher grade
Predictive	Predict response to specific therapies (e.g., RT, ADT, systemic therapy)	Decipher [30], PORTOS [30], GPS [24], AR-V7 [31,33,37]	Tissue, CTCs	Treatment intensification or de-intensification	Post-surgery, high-risk or ADT-exposed	Decipher: guides adjuvant RT PORTOS: radiosensitivity profile GPS: may predict treatment benefit AR-V7: resistance to AR-targeted therapies
Surrogate	Indicate early treatment response or progression before clinical endpoints	PSA kinetics [34], SUVmax [9], ADC [35]	Blood, imaging	Monitor treatment efficacy or failure	Follow-up, clinical trials	PSA kinetics: doubling time, nadir, biochemical recurrence SUVmax: tumor burden ADC: high cellularity/aggressiveness
Theranostic	Combine diagnostic and therapeutic utility via molecular targeting	PSMA PET [36]	Imaging	Staging and focal therapy planning	Intermediate/high-risk localized disease	PSMA PET: visual and SUV-based PSMA expression for staging and biologically guided treatment

**Table 2 ijms-26-07667-t002:** Clinical use, interpretation, and validated setting of biomarkers in localized prostate cancer.

Use	Test	Biomarker	Sample	AUC/NPV	Scoring and Interpretation	Advantages	Limitations	Validated Clinical Setting
Avoid initial and subsequent biopsies	PHI [15,28,39]	PSA, free PSA, isoform [−2] proPSA	Blood	AUC: 0.70–0.75	Score: 0–55 Risk >40 associated with significant PCa PHI > 55: 50% chance of PCa	Accessible and fast. Higher sensitivity and specificity than PSA, detects high-risk PCa. Complementary to PSA in AS to detect biochemical progression.	Lower sensitivity in small tumors.	Initial evaluation with PSA 4–10 ng/mL.
	4K Score [40,41]	PSA, free PSA, intact PSA, hK2 + rectal examen, age, and previous biopsy	Blood	AUC: 0.82–0.87 NPV: 95%	Score: 0–100; risk of Gleason ≥7 PCa	Integrates clinical variables, high precision in high-risk PCa.	High cost, not always available.	Patient selection for initial biopsy.
	Stockholm3 [19,42]	PSA + 232 SNPs + 6 plasmatic proteins	Blood	AUC: 0.81–0.85	Score: 0–15 >11 suggests significant PCa	Includes genetic risk, avoids 50% of biopsies.	Only available in Europe.	Screening for the general population.
	SelectMDx [20,43,44]	mRNA from HOXC6, TDRD1 and DLX1 genes.	Urine post-DRE	AUC: 0.76 NPV: 90%	Score 0–1: positive = high risk of significant PCa	Identifies high-risk PCa. Better in combination with mpMRI.	Limited availability, influenced by sample gathering.	Decision to perform biopsy after high PSA.
	ExoDX [22,50,51]	Exosomal RNA from PCA3, ERG, and SPEDF	Urine (no DRE)	AUC: 0.71–0.75	Continuous score; >15.6 is threshold for biopsy	No DRE required, useful after PSA or mpMRI.	Limited use outside the United States.	Pre-biopsy. PSA 2–10 ng/mL.
	MiPs [49]	PCA3 + PSA and TMPRSS2-ERG/ETV	Urine post-DRE	AUC: 0.77–0.81 NPV: >90%	Individual risk; the higher the score, the higher the risk	Improves the identification of high-risk PCa (better than only PCA).	Low specificity, requires DRE, limited evidence in some populations.	PSA 2–10 ng/mL with no previous biopsy.
Re-biopsy	PCA3 [45,46,47,48]	Non-coding mRNA PCA3	Urine post-DRE	AUC: 0.66	Continuous score; >35 means higher risk of PCa	Not affected by prostatic volume. Better predictor of PSA.	Only useful if combined with mpMRI. Outdated by more precise tests.	Patients with a previously negative biopsy.
	ConfirmMDx [52,53,110,113]	DNA methylation in APC, RASSF, and GSTP1	Tissue	AUC: 0.76 NPV: 88–96%	Binary result (positive/negative) for methylation	High NPV (>90%) after negative biopsy. Detects the halo effect.	Only applicable after previous biopsy. Not useful in inflammation High cost.	Decision to re-biopsy after a previously negative result.
Indication/ exclusion of AS	Oncotype Dx [25,62,63]	17 genes (proliferation, invasion…)	Tissue	AUC: 0.68–0.72	Score 0–100; >40 means increased risk of progression	Reclassifies Gleason 6–7. Predicts upgrading and progression. Useful in candidates for AS.	Cost. Requires solid sample.	Choice for AS if Gleason ≤ 7.
	Prolaris [64,65,66,82]	31 cell cycle genes and 15 maintenance genes	Tissue	AUC: 0.77–0.88	Score: 0–10 CCP >1: higher risk of progression	Robust data, easy to interpretate. Clear stratification for low risk.	Not tailored for high-risk disease. The interpretation requires experience.	Decision for AS if low Gleason score with rising PSA.
	Decipher [57,58,59,60,61]	RNA from 22 genes (metastasis); GPS	Tissue	AUC: 0.75–0.80	Score: 0–1 >0.6: high risk <0.4: low risk 0.4–0.6: intermediate risk	Good predictor in Gleason 7–8. High prognostic discrimination.	Cost.	Exclusion for AS; risk of early metastasis.
	Promark [27,69]	Proteomic signature of 8 proteins associated with tumor aggressiveness	Tissue	AUC: 0.70–0.78	Score: 0–1 (continuous) >0.33 increasing risk of progression or upgrading; >0.8: high risk (77% Gleason > 4+3 o T3+)	Does not require complex techniques, useful in Gleason 3+3 and 3+4.	Only applicable in tissue; less validated than Decipher/Oncotype.	Choice for AS if Gleason 3+3 and 3+4.
Treatment intensification	Decipher [29,84,85,86,114]	RNA from 22 genes; GC score	Tissue	AUC: 0.77	Score: 0–1; >0.6: high risk of metastasis	Robust stratification after prostatectomy. Predicts the risk of metastasis, recurrence, and mortality. Guides the use of ADT after RT.	Requires enough tissue. Cost. Limited prospective validation.	Post-prostatectomy with + margins or pT3. Salvage RT. Intermediate/high risk. Guides adjuvant ADT.
	Prolaris [81,82]	31 cell cycle genes	Tissue	AUC: 0.77–0.88	Continuous score; CCR >1: higher risk of progression	Observational data, long follow-up. Supports the decision for treatment intensification.	Not useful if ADT is already necessary. Lower impact in high risk.	Pretreatment in intermediate risk. ADT indication unclear.
	Oncotype Dx [24,83]	17 genes (proliferation, invasion…)	Tissue	AUC: 0.68–0.72	Score: 0–100; >40 means high risk of progression or upgrading	Stratifies Gleason 6–7. Identifies candidates for intensification in intermediate risk.	No estimation of long-term metastasis. Limited post-operatory validation.	Gleason 6–7 pretreatment. Intermediate risk. Decision between AS VA and intensified treatment.

A summary of biomarkers used across diagnostic, prognostic, and therapeutic scenarios in localized prostate cancer. The table details the sample type, area under the curve (AUC) or negative predictive value (NPV), scoring system, clinical advantages and limitations, and the validated setting for each biomarker. Biomarkers are categorized by use: biopsy decision, active surveillance eligibility, and treatment intensification or de-intensification. Abbreviations: ADT = androgen deprivation therapy; AS = active surveillance; AUC = area under the curve; CCP = cell cycle progression; CTCs = circulating tumor cells; DRE = digital rectal examination; GPS = genomic prostate score; mpMRI = multiparametric magnetic resonance imaging; mRNA = messenger RNA; NPV = negative predictive value; PCa = prostate cancer; PHI = prostate health index; PSA = prostate-specific antigen; RT = radiotherapy; SNPs = single-nucleotide polymorphisms.

## Data Availability

Data sharing not applicable. No new data were created or analyzed in this study.

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
