# Peer review of "Biomarkers in Localized Prostate Cancer: From Diagnosis to Treatment"

_ijms, 2025, doi:10.3390/ijms26167667_

Round 1

Reviewer 1 Report

Comments and Suggestions for Authors

Title - localized cancer is not the main topic of the manuscript - Major

abstract - concise review of the manuscript - No remarks

Introduction - short and poorly structured - Major

Material and Methods - should be more detailed - period of article search, method of choosing the most relevant research and resolving disputes, responsibilities of different authors for different chapters, etc. - Major

Discussion and conclusion - vague and theoretical, should be more focused - Major 

Author Response

Comment 1:
Title – “localized cancer” is not the main topic of the manuscript – Major

Response 1:
Thank you for this observation. We have reviewed the title to ensure it accurately reflects the clinical focus of the manuscript, which specifically addresses the use of biomarkers in localized prostate cancer. Since the entire article is structured around diagnostic and therapeutic strategies in localized settings (both de novo and post-treatment recurrence), we believe it is appropriate to retain the reference to “localized prostate cancer” in the title. Nevertheless, we have reinforced this thematic focus throughout the manuscript.
Revised in: Title and throughout the manuscript text.

Comment 2:
Introduction – short and poorly structured – Major

Response 2:
We fully agree and have rewritten the introduction to include a more comprehensive background, a clearer rationale for the study, and a more structured outline. The revised section now highlights the clinical challenge posed by localized prostate cancer and the potential role of biomarkers across different clinical scenarios.
Revised in: Page 2, paragraphs 1 and 2.

Comment 3:
Material and Methods – should be more detailed: period of article search, method of choosing the most relevant research and resolving disputes, responsibilities of different authors for different chapters, etc. – Major

Response 3:
Thank you for this valuable suggestion. The Methods section has been significantly expanded to include:

  • Literature search period (January 2000 to June 2025)
  • Inclusion and exclusion criteria
  • Dual review process with conflict resolution by consensus
  • Assignment of specific sections to co-authors based on clinical expertise (urology, medical oncology, radiation oncology)
    Revised in: Page 4, paragraphs 1–2.

Comment 4:
Discussion and Conclusion – vague and theoretical, should be more focused – Major

Response 4:
We appreciate this insightful comment. The Discussion section has been extensively revised and reorganized around three key clinical questions:

  1. How to identify patients who truly need biopsy?
  2. How to distinguish indolent from aggressive tumors?
  3. How to tailor treatment choice in patients with localized prostate cancer?

To support this practical approach, we added two visual clinical algorithms (diagnostic and therapeutic) and repositioned our detailed summary table (Table 2) within the Discussion for immediate clinical applicability. The Conclusion has also been rewritten to provide a concise and evidence-based summary.
Revised in: Pages 17–21 (Discussion and Conclusion),
Figures 1 and 2 (Diagnostic and Therapeutic Algorithms),
Table 2 (now embedded in Discussion section).

Response to Comments on the Quality of English Language

Point 1:
No specific comment was provided, but the reviewer implied a need for better clarity and structure.

Response 1:
We have conducted a full language review with the assistance of a native English editor. Grammatical improvements, clarity adjustments, and the elimination of redundant expressions have been applied throughout the manuscript. All edits are visible in the tracked changes version.

Reviewer 2 Report

Comments and Suggestions for Authors

The article offers a concise overview of current developments in biomarker research aimed at improving the clinical management of localized prostate cancer (PCa). Traditional reliance on Prostate-Specific Antigen (PSA) is rightfully criticized for its limited specificity and prognostic power, leading to the exploration of novel biomarkers and imaging tools, such as urinary assays (PCA3, SelectMDx), tissue-based genomic panels (Oncotype DX, Prolaris, Decipher), and advanced imaging modalities (mpMRI, PSMA-PET). The manuscript written in a form of narrative review, addresses a critical issue in oncology—personalizing PCa diagnosis and treatment through more accurate, molecular-based tools. It includes a spectrum of biomarker types (blood, imaging), reflecting the interdisciplinary nature of modern PCa research. While the article successfully identifies the limitations of PSA and summarizes promising alternatives, it ultimately falls short in delivering a comprehensive, practical, and analytically rigorous review of modern biomarkers in localized prostate cancer. For a paper positioned within the domain of personalized medicine many critical remarks are present:

Major Remarks

  • Lack of Conceptual Clarity on Biomarker Categories: The article fails to define and classify the various types of biomarkers, which is foundational for any meaningful clinical application. It omits crucial distinctions such as:
    • diagnostic biomarkers (e.g., PSA, PCA3)
    • prognostic biomarkers (e.g., Decipher)
    • predictive biomarkers (tumor appearance or)
    • surrogate biomarkers (treatment response)
    • theranostic biomarkers (therapy)

Without this context, readers are left without a framework to understand which biomarker serves what clinical role.

-Absence of Methodological Detail

The article does not explain:

    • How each biomarker is calculated
    • Why it was introduced
    • What makes it novel or better than existing tools
    • What is recommended as first-line vs second-line testing

For instance, mentioning “SelectMDx” without explaining its basis in mRNA quantification of DLX1 and HOXC6 or its use in identifying patients for biopsy avoidance leaves an informational gap.

-No Diagnostic or Clinical Algorithm

There is no attempt to synthesize the reviewed biomarkers into a coherent diagnostic or treatment algorithm. As a result, the article does not provide a usable clinical pathway or “take-home” message for practitioners. For a paper advocating personalized medicine, the lack of algorithmic guidance is a major shortcoming.

-Superficial Treatment of Artificial Intelligence (AI)

AI is only briefly mentioned, without any elaboration. The article misses the opportunity to explore:

  • Existing AI tools for radiomics or digital pathology
    • AI-based risk prediction models combining clinical and genomic data
    • The potential of machine learning to stratify patients for active surveillance vs intervention

In order to improve and to elevate the paper from a descriptive overview to a clinically actionable and conceptually grounded resource, it should

-include a tabulated summary of biomarker types, definitions, clinical utility, and performance metrics.

-propose evidence-based diagnostic and treatment algorithms.

-prepare AI for predictive, diagnostic, surogate and theranostic models.

-New references are failed. Springer monograph contains article: Biological markers of therapeutic response in prostate cancer in a book: Prostate Cancer: Advancements in the Pathogenesis, Diagnosis and Personalized Therapy ISBN 978-3-031-51711-2;  DOI : 10.1007/978-3-031-51712-9. Springer

Comments on the Quality of English Language

no comment

Author Response

Comment 1 – Lack of Conceptual Clarity on Biomarker Categories:
The article fails to define and classify the various types of biomarkers, omitting key distinctions such as diagnostic, prognostic, predictive, surrogate, and theranostic biomarkers.

Response 1:
Thank you for this valuable suggestion. We fully agree. We have added a dedicated section (Section 3: “Classification and Clinical Roles of Biomarkers”) where biomarkers are categorized into five functional classes: diagnostic, prognostic, predictive, surrogate, and theranostic. Each category is defined and illustrated with clinical examples and rationale. Additionally, Figure 1 maps these biomarker types onto the clinical workflow, while Table 1 provides a structured overview aligned with their functional classification.
See revised manuscript: Section 3 (pages 4–7), Figure 1, and Table 1.

Comment 2 – Absence of Methodological Detail:
The article does not explain how biomarkers are calculated, what makes them novel, or how they are positioned in clinical pathways.

Response 2:
We agree and have revised the manuscript accordingly. Detailed explanations have been added regarding:

  • Calculation methods or scoring algorithms (e.g., PHI formula, 4Kscore, Decipher risk score);
  • Molecular or biological rationale underlying each test;
  • Clinical position in first-line or second-line decision-making (e.g., SelectMDx, ExoDx).
    These clarifications appear both in the main text and in the updated tables.
    See revised manuscript: Section 3 (pages 5–8) and Table 1.

Comment 3 – No Diagnostic or Clinical Algorithm:
There is no attempt to synthesize biomarkers into a coherent diagnostic or treatment algorithm.

Response 3:
We appreciate this important comment. We have now included Figure 2, a comprehensive diagnostic and therapeutic algorithm for biomarker-guided decision-making in localized prostate cancer. It integrates PSA, mpMRI, molecular tests, and risk classifiers to inform biopsy decisions, surveillance, and treatment intensification—including biochemical recurrence management.
See revised manuscript: Figure 2, Section 6 (pages 15–16), and Discussion (pages 24–25).

Comment 4 – Superficial Treatment of Artificial Intelligence (AI):
AI is only briefly mentioned. The paper should discuss AI tools for radiomics/pathology, risk models, and machine learning applications.

Response 4:
We agree and have significantly expanded the discussion in Section 7: “Artificial Intelligence Applications in Prostate Cancer Radiomics”. It now covers:

  • AI tools in radiomics using mpMRI and PSMA-PET (e.g., RadClip, RadS);
  • AI-based prognostic models such as ArteraAI;
  • Machine learning applications for treatment response prediction and surveillance selection;
  • Associated performance metrics (AUC, HR) and validation studies.
    See revised manuscript: Section 7 (pages 20–23).

Comment 5 – Include a tabulated summary of biomarker types, definitions, utility, and metrics:

Response 5:
This has been fully addressed through the inclusion of two detailed summary tables:

  • Table 1: Functional classification of biomarkers by clinical role;
  • Table 2: Clinical utility, scoring system, interpretation, performance (AUC, NPV), advantages, limitations, and validated context.
    See pages 8–14 and 27–28 of the revised manuscript.

Comment 6 – Propose evidence-based diagnostic and treatment algorithms:

Response 6:
As noted above, Figure 2 presents a clinical workflow from PSA testing to advanced biomarker-guided therapy. Each decision node incorporates evidence-based thresholds and validated tools.
See revised manuscript: Figure 2, Section 6 (pages 15–16), and Discussion (pages 24–25).

Comment 7 – Prepare AI section for predictive, diagnostic, surrogate, and theranostic models:

Response 7:
We have expanded the AI section accordingly. For each biomarker role—diagnostic, predictive, surrogate, and theranostic—we now describe at least one AI application and its clinical performance.
See revised manuscript: Section 7 (pages 20–23).

Comment 8 – Missing references, including Springer monograph (DOI: 10.1007/978-3-031-51712-9):

Response 8:
Thank you. We have now included the Springer monograph as Reference 14, and integrated its content in the discussion of emerging biomarkers and therapeutic response.
See revised manuscript: Bibliography (page 16) and Reference 14.

Response to Comments on the Quality of English Language

Point 1: General improvement recommended.
Response 1: The manuscript has been fully edited by a native English speaker with experience in biomedical writing. Grammar, syntax, and style have been improved throughout, with all changes marked in track changes.

Round 2

Reviewer 1 Report

Comments and Suggestions for Authors

extensively re-written and significantly improved manuscript

few additional points for review

fig 1 -  Biopsy - or equivocal mpRMI  - typo - MRI? - Minor

 7. Artificial Intelligence Applications in Prostate Cancer Radiomics - it seems like this section has been added due to other reviewer`s recommendation, but in this reviewer`s opinion radiomics is outside the context of the manuscript - Minor

 8. Future perspectives - this paragraph may be situated after Discussion and before Conclusions - Minor

Author Response

  1. Point-by-point response to Comments and Suggestions for Authors

Comment 1: “fig 1 - Biopsy - or equivocal mpRMI - typo - MRI? - Minor”

Response 1: Thank you for spotting this typographical error. We have corrected “mpRMI” to “mpMRI” in Figure 1 and the corresponding caption.
Location in revised manuscript: Figure 1 caption and figure content.
“Corrected label: ‘Biopsy or equivocal mpMRI’”

Comment 2: “7. Artificial Intelligence Applications in Prostate Cancer Radiomics - it seems like this section has been added due to other reviewers recommendation, but in this reviewers opinion radiomics is outside the context of the manuscript - Minor”

Response 2: We appreciate the reviewer’s thoughtful observation. This section was indeed expanded following previous reviewer suggestions. However, we agree that certain radiomic details may have been too technical.
We have now streamlined the section to better align with the manuscript’s clinical focus, emphasizing diagnostic and prognostic applications of AI relevant to localized prostate cancer and removing excessive technical details.

Location in revised manuscript: Section 7, pages 23–25 (paragraphs 2–3).

Comment 3: “8. Future perspectives - this paragraph may be situated after Discussion and before Conclusions - Minor”

Response 3: Thank you for this suggestion. We agree with the recommendation and have moved the “Future Perspectives” section (Section 9) to follow immediately after the Discussion (now Section 8) and before the Conclusions.

Location in revised manuscript: New order—Section 8: Discussion, Section 9: Future

  1. Response to Comments on the Quality of English Language

Point 1: “The English is fine and does not require any improvement.”

Response 1: We thank the reviewer for this positive assessment of the language quality. No further edits were required in this regard.

Reviewer 2 Report

Comments and Suggestions for Authors

The authors have carefully considered  the reviewers comments and suggestions. In this way, they have made a concerted effort to incorporate recent advances in the diagnosis of PCa, providing an updated and practical perspective. In light of current evidence, the manuscript now offers practical guidelines and a refined classification of biomarkers, aiming to enhance their clinical applicability and interpretability. This structured approach contributes to the translational value of the work and supports its relevance for both research and clinical practice.

Author Response

Point-by-point Response to Comments and Suggestions for Authors

Comment 1:
The authors have carefully considered the reviewers’ comments and suggestions. In this way, they have made a concerted effort to incorporate recent advances in the diagnosis of PCa, providing an updated and practical perspective. In light of current evidence, the manuscript now offers practical guidelines and a refined classification of biomarkers, aiming to enhance their clinical applicability and interpretability. This structured approach contributes to the translational value of the work and supports its relevance for both research and clinical practice.

Response 1:
We sincerely thank you for your positive assessment and recognition of the effort to incorporate recent developments and structure the content in a clinically useful format. We are pleased to know that the updated classification and practical guidelines were considered valuable and translationally relevant. Our goal was precisely to bridge scientific knowledge with clinical practice, and your comments affirm that direction. No further changes were deemed necessary in response to this comment.

  1. Response to Comments on the Quality of English Language

Point 1:
The English is fine and does not require any improvement.

Response 1:
Thank you for your evaluation. We have nonetheless reviewed the entire manuscript once again to ensure consistency, clarity, and style.
